# Extensible membrane nanotubules mediate attachment of *Trypanosoma cruzi* epimastigotes under flow

Cristhian David Perdomo-Gómez[1,2], Nancy E. Ruiz-Uribe[2,3], John Mario González[1], Manu Forero-Shelton[3]*

1 Laboratorio de Ciencias Básicas Médicas, School of Medicine, Universidad de los Andes, Bogotá, Colombia, 2 Department of Biological Sciences, Universidad de los Andes, Bogotá, Colombia, 3 Department of Physics, Universidad de los Andes, Bogotá, Colombia

* anforero@uniandes.edu.co

**Data Availability Statement:** The datasets analyzed during the current study are available on Figshare (https://doi.org/10.6084/m9.figshare.22178795).

## Abstract

*Trypanosoma cruzi* is the etiological agent of Chagas disease, an important cause of infectious chronic myocardiopathy in Latin America. The life cycle of the parasite involves two main hosts: a triatomine (arthropod hematophagous vector) and a mammal. Epimastigotes are flagellated forms inside the triatomine gut; they mature in its intestine into metacyclic trypomastigotes, the infective form for humans. Parasites attach despite the shear stress generated by fluid flow in the intestines of the host, but little is known about the mechanisms that stabilize attachment in these conditions. Here, we describe the effect of varying levels of shear stress on attached *T. cruzi* epimastigotes using a parallel plate flow chamber. When flow is applied, parasites are partially dragged but maintain a connection to the surface via ~40 nm wide filaments (nanotubules) and the activity of flagella is reduced. When flow stops, parasites return near their original position and flagellar motion resumes. Nanotubule elongation increases with increasing shear stress and is consistent with a model of membrane tether extension under force. Fluorescent probes used to confirm membrane composition also show micron-wide anchoring pads at the distal end of the nanotubules. Multiple tethering accounts for more resistance to large shear stresses and for reduced flagellar movement when flow is stopped. The formation of membrane nanotubules is a possible mechanism to enhance adherence to host cells under shear stress, favoring the continuity of the parasite´s life cycle.

## Introduction

*Trypanosoma cruzi*, a flagellated protozoa, is the etiological agent of Chagas disease, a condition that afflicts mainly regions in Latin America [1–3]. The life cycle of the pathogen is complex, consisting of three parasitic forms: amastigote, epimastigote, and trypomastigote [1,2,4]; these can be differentiated according to the presence or absence of an external flagellum, in addition to the position of the kinetoplast (extranuclear DNA organelle) relative to the nucleus [4,5].

The flagellum is one of the most characteristic organelles in *T. cruzi*. It emerges from the basal body, through the flagellar pocket which is an invagination responsible for exocytosis

**Funding:** MF received support from the Physics Department and the Vice-presidency of Research for the use of the microscopy facility (µ-Core), and support for materials and reagents from the Physics Department. JMG received support from the School of Medicine for materials and reagents. CP was partially supported from a "programa" of the faculty of sciences. The specific roles of these authors are articulated in the 'author contributions' section. The authors also acknowledge support from "fondos de la convocatoria de la Vicerrectoría de Investigación y Creación" for partial financing of the publication fees. The funders had no role in study design, data collection and analysis, decision to publish, or preparation of the manuscript.

**Competing interests:** The authors have declared that no competing interests exist.

and endocytosis involved in host-parasite interaction [6,7]. The flagellum is involved in morphogenesis, cell division [8,9], evasion of the immune system, by mobilizing surface bound antibodies to the flagellar pocket, where those are transported to the lysosome to be degraded [10], and motility through the different anatomical sites and compartments of both the vector and the mammalian host [11–13].

Adhesion to surfaces is a crucial step for replication and maturation, the process of producing the infective form of the parasite in the vector's gut [14]. The flagellar membrane plays a fundamental role in the initial process of attachment of the parasites to surfaces [15] and it has been shown that parasites adhere to surfaces by the anterior tip of the flagellum [15]. Most studies of *T. cruzi* invasion have been focused on the characterization of the molecules involved in the adhesion/invasion process [16], and less attention has been paid to the adaptability of these parasites to variable, hostile and confined microenvironments inside the hosts, which is a challenge for the survival of the pathogen. Fluids present in the host microenvironment can play a defensive role, preventing effective colonization [17,18]. For example, human arterial circulation has shear stresses approximately between 10 and 70 Dyn/cm$^2$ (1–7 Pa); while in venous circulation this magnitude oscillates between 1 and 6 Dyn/cm$^2$ (0.1–0.6 Pa) [19], a hostile environment for hematic parasites. However, studies on the mechanics of adhesion of *T. cruzi* to surfaces under shear stress, either in the intestine or heart, are absent.

Here, in order to understand how *T. cruzi* epimastigotes attach under flow inside the host, we performed parallel-plate flow chamber experiments with time-varying shear stresses as well as several types of microscopy to study the mechanical response to fluid flow by the membranes of flagella and propose a new role in the adhesion process.

## Methods

### Culture and maintenance *of T. cruzi* epimastigotes

*T. cruzi* epimastigotes from DA isolate (MHOM/CO/01/DA) a TcI genotype were used. Parasite were cultured in LIT (Liver Infusion Tryptose) medium supplemented with 5% bovine fetal serum (Eurobio, Les Ulis, France), and penicillin/streptomycin solution was added to prevent bacterial contamination. From the culture, parasites were diluted with the dye trypan blue (Sigma-Aldrich, Saint Louis, MI, USA) to count them in a Neubauer chamber. Epimastigotes were transferred to 4 mL tubes, in which 3 mL of 0.01 M phosphate buffer saline pH 7,4 (PBS 1X, Sigma-Aldrich) was added. Then, each tube was centrifuged at 1350 g for 5 min to remove excess culture medium. Two washings per sample were carried out, and the whole procedure was repeated twice. Finally, each sample was resuspended in 1 mL of PBS 1X, ensuring a concentration of 1 million epimastigotes per milliliter.

### Epimastigote fluorescent staining

We used Carboxy fluorescein succinimidyl ester (CFSE), a cytoplasmic dye which covalently adheres to intracellular amine residues, and Octadecyl Rhodamine B Chloride (R18), a lipophilic cationic dye derived from rhodamine as a membrane dye. At least 1 µL of CFSE and 1 µL of R18 (250 µg/mL, ethanol) were added per every million epimastigotes contained in a 1.5 mL conic tube. The tube was covered aluminum foil, incubated in dark at 37°C for 15 min and mixed gently every 3 min. Labeled parasites were washed five times in PBS 1x. For CFSE, fluorescence was assessed by flow cytometry with 510/20 PMT comparing with unstained parasite; R18 fluorescence was assessed by fluorescence microscopy, using a TRITC fluorescence cube (excitation filter 528–553 nm, dichroic: 565 nm, and emission filter: 590–650 nm). To check for fluorescence crossover, epimastigotes dyed with only CFSE were tested in the fluorescence microscope using a TRITC fluorescence cube, and no excitation of the dye was

observed using the same exposure times, and camera settings as used for taking data for the images. Similarly, epimastigotes dyed only with R18 were tested with a FITC cube (excitation filter 465–495 nm, dichroic: 505 nm, and emission filter: 515–555 nm), and no excitation was observed; thus, fluorescence crossover was discarded.

To quantify fluorescence intensity of CFSE and R18, ImageJ [20] software was used. Intensity measurements were taken at the nanotubule, the anchoring pad and the background of each image. The background value was subtracted from the nanotubule and pad measures, and the intensity ratio between the pad and the nanotubule was determined. This procedure was carried out for two different stained parasites and the average measure was used.

## Microfluidic flow chamber experiments

A Glycotech (Gaithersburg, MD, USA) microfluidic chamber (product # 31–001) was used. The silicone gasket used was that corresponding to the channel of 0.25 cm wide and 0.0254 cm high. This system was sealed with either a 35 mm Corning (New York, NY, USA) plastic dish, or a # 1.5 round glass coverslip 45 mm in diameter when short working distance objectives were used, for high resolution or fluorescence. Coverslips were not functionalized for simplicity since good adhesion was observed, although good attachment could also be obtained with poly-l-lysine functionalized coverslips. The chamber was connected to a vacuum pump (Barnant-Company vacuum pump reference 400–3910), to ensure a proper seal of the gasket.

A three-way stopcock was used to select epimastigotes or PBS 1X flow. A 1 mL plastic syringe (Nubenco, Paramus, NJ, USA) was used to dispense the epimastigotes. A 60 mL Becton-Dickinson (BD, Queretaro, Mexico) plastic syringe was used to dispense the PBS 1X; its flow was controlled by a syringe pump (Pump 11 Elite Infusion / Withdrawal Programmable Single Syringe, Harvard Apparatus, Inc.) varying from 0.5 to 2.5 mL/min (0.0083–0.0417 cm$^3$/s). Eq 1, which was provided by the manufacturer, was used to calculate shear stress in the parallel flow chamber; results are obtained in Pa, but we use pN/μm$^2$ which are equivalent. Equivalences of PBS 1X flow to shear stress are presented in S1 Table.

$$\tau = \frac{6\mu Q}{h^2 W} \tag{1}$$

**Eq 1** Shear stress as a function of flow rate and chamber size. Here μ is the fluid viscosity (μ = 0.001 Pa·s for PBS), Q the fluid flow in cm$^3$/s, h the channel height in cm and W the channel width in cm.

## Optical microscopy

Observations with phase contrast and Differential Interference Contrast (DIC) microscopy were made on two systems: one, an inverted Zeiss Axiovert 40 CFL microscope, with Zeiss A-Plan 40X / 0.50 *Ph*2 and Zeiss A-Plan 100X / 1.25 Oil objectives and Point-Grey GRASS-HOPPER 3 camera. Micro-Manager [21] software was used for image acquisition. Images were also obtained with a Nikon Eclipse Ti microscope using the Plan Fluor 40x Oil DIC H N2 and the Apo TIRF 60x Oil DIC N2 objectives with 1.49 NA and DIC. 90x magnification was obtained by using the 60x objective and a magnification changer of 1.5. Videos were taken with an ORCA-R2 (Hamamatsu Photonics K.K., Systems division) camera, while fluorescence images were taken with an Andor ixon 897 (Andor Technology, Oxford Instruments) camera with EM gain set to 0. at 16 bits in order to get the dynamic range necessary to see both the bright and faint features of the epimastigotes, using an exposure time of 300 ms, and a cooling temperature of -60˚C.

Videos were taken at 10 frames per second, with 100 ms exposure time. Image parameters after video acquisition, such as contrast, were adjusted with ImageJ [20] software for better presentation on computer screens.

Flagellated forms of *T. cruzi* move towards the flagellum, and by following individual parasites on the microscopes it was possible to turn on the shear stress at the moment of initial adhesion to the surface via the anterior tip of flagellum. The structure of flagella did not change, and tracking was achieved by following the position of its anterior tip using ImageJ software with the MTrackJ complement. The position data was obtained as two-dimensional coordinates, in pixels and calibrated using a calibration ruler and the calibration feature of imageJ.

## Fabrication of microchannels with PDMS

Due to the short working distance of fluorescence objectives, we fabricated microchannels with polydimethylsiloxane (PDMS) over ~170 μm thick (#1.5) rectangular glass coverslips. In a plastic container, curing agent and PDMS were mixed, in a 1:10 proportion, with a plastic fork for 5 min. The mixture was left in a degasser for at least two hours and poured over a channel mold 0.25 cm wide, 2 cm long and ~340 μm tall (two cut coverslips bonded with optical glue), and left in the degasser for another two hours, or until the air bubbles disappeared. Then, the cured PDMS was heated to 65°C in an incubator overnight.

After solidification, the PDMS was unmolded and cut with a blade in order to fit it on a coverslip, and holes at the longitudinal ends of the channel (approximately 2 mm from the edge) were made with a 3/4 mm perforator. The PDMS was immersed in isopropanol and sonicated for 40 min, dried with filtrated pressure air and then incubated at 65°C overnight. The coverslips were immersed in 1M KOH, sonicated for 30 min, washed 5 times with ultrapure water, dried with filtrated pressure air, and incubated at 65°C overnight.

PDMS (channel face up) and coverslips received oxygen plasma treatment for 1 min using a plasma cleaner PDC-32G (Harrick Plasma, Ithaca, NY, USA) in order to activate both surfaces; then, they were assembled and left on a hot plate at 150° C for 5 min, coverslip facing down. Finally, they were incubated overnight at room temperature.

## Scanning electron microscope (SEM)

Epimastigotes were dispensed in the microfluidic chamber and subjected to flow during a period of at least 7 min after the formation of nanotubules, because it was observed that was the time necessary to ensure the nanotubules remained extended (likely due to additional attachments forming as explained in the results section). Once stretched, the parasites were incubated overnight at room temperature with a 2.5% glutaraldehyde solution to fix them to the surface; the chamber was disassembled, maintaining the surface of the coverslip where the epimastigotes were located humidified with glutaraldehyde. Then, epimastigotes were dehydrated by ethanol dilution series and then critical-point dried. A Denton Vacuum Desk IV Standard Sputter Coater was used to cover the samples with a gold layer ~10 nm thick, similar to what was used in [22]. Electron images were obtained in a Lyra3 Tescan, at a 5.0 kV accelerating voltage and working distance of 7.89 mm.

## Analysis of data and statistics

Data was exported to Excel (Microsoft, Redmond, WA, USA). Pythagoras theorem was used to find the length of the nanotubule stretching for each epimastigote. Pixel size was calibrated with a micro ruler in ImageJ.

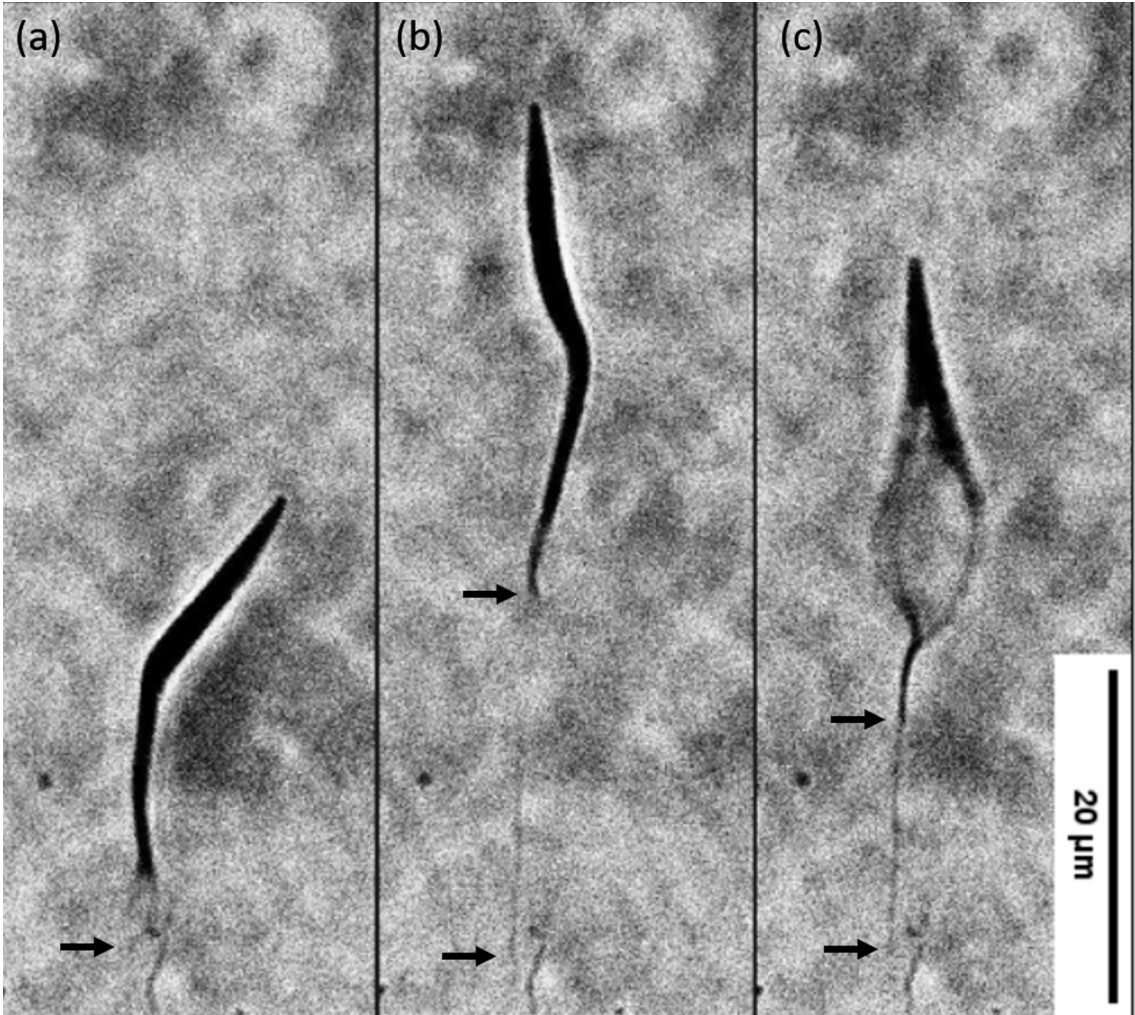

**Fig 1. Filament formation.** (a) Surface-bound epimastigote, at 0 shear stress. (b) The same epimastigote under a shear stress of 0.62 pN/μm². A tether-like structure emerges from the flagellum, the arrows delimitate its length. (c) The flagellum resumes its beating motion while the tether is still attached to the surface. In this case the parasite did not return to its initial position.

The force threshold required for nanotubule extrusion was estimated from the intercept of the rapid elongation phase presented in Fig 1, by calculating the stokes drag using the formula by Goldman for a sphere [23], with a particle radius R = 1μm, given the diameter of 2 μm for epimastigotes and not taking into account its length.

The bending modulus was calculated using the formula derived by Hochmuth et al. [24] and Brochard-Wyart et al. [25]

$$\kappa = \frac{f_0 \cdot r_t}{2\pi} \tag{2}$$

**Eq 2** Bending modulus ($\kappa$) as a function of threshold pulling force $f_0$ and tether radius $r_t$.

Surface viscosity $\eta_e$ was obtained from Eq (16) in Brochard-Wyart et al. [25] using the elongation velocity $\dot{L}$ vs force curve (S1 Fig); data was fit to the following equation using minimum squares and assuming a bond density $v$ between the cytoskeleton and the membrane of $10^3$/

$\mu m^2$, $\kappa$ as estimated above and $r_t$ from the electron microscopy data:

$$f(\dot{L}) = 2\pi \left( 2\kappa^2 \nu \eta_e \dot{L} \ln\left(\frac{R}{r_t}\right) \right)^{1/3}$$

(3)

Eq 3 Extrusion force as a function of elongation velocity $f(\dot{L})$.

## Results

### Attached epimastigotes extend significantly in response to shear stress

Epimastigotes bound nonspecifically to the surface of the flow cell were subjected to different fluid flows of 1X PBS and followed using phase contrast optical microscopy. The parasites responded in one of two manners under flow: some barely shifted their position while others moved significantly from their initial position, up to 10's of microns, returning near, or to their starting position. We first focus on the group that changed position when shear stress was applied, which were obtained in a systematic manner by following individual parasites under the microscope until they attached and immediately turning on fluid flow. Fluid flow flipped the parasites (which travel flagellum-first) around the tip of the flagellum (Fig 1A), and at higher flows those that remained attached formed thin filaments that generally emerged from the flagellar tip and extended from there to the initial position of the parasite (Fig 1B). Significant extension of the soma of the parasites was not observed, so in this study parasite extension will refer to that of the filaments.

### Nanotubules connect an anchoring pad to the flagellum

To identify the structural changes and the structures involved in epimastigote extension, surface-bound epimastigotes were observed at 100x magnification (Fig 1) starting at no flow (Fig 1A) and subjected to a shear stress of 0.62 pN/$\mu m^2$. Under the optical microscope, filaments appear to be ~200nm wide, the resolution limit of the transmitted light microscope, and of variable length, up to several times the length of the epimastigote (100 $\mu m$). Notably, during flow, parasite movement stopped and resumed when flow stopped (Fig 1C and S1 Movie) suggesting there is a sensing mechanism that stops movement when filaments extend.

Scanning electron microscopy (SEM) was used to visualize the filamentous structure at high resolution. Fig 2 shows an epimastigote after being stretched with a shear stress of 1.24 pN/$\mu m^2$, immediately fixed, then prepared for SEM imaging by dehydration and gold sputtering. In Fig 2A, a nanotubule—as we will refer to these structures from here on, extends ~2 $\mu m$ from the tip of the flagellum of the epimastigote. The nanotubule is ~60 nm wide (Fig 2B) and connects to a pad at its distal end. Since the gold is ~10 nm thick, the diameter of the nanotubule is estimated at ~40 nm. Near the flagellum, the neck of the nanotubule is funnel shaped, as wide as the flagellum tip (~130 nm) on one side, and as wide as the nanotubule on the other. The main section of the nanotubule is of constant width, and its distal end broadens to form a pad about a micron long and several hundred of nm wide. This structure was not initially observed in transmitted light microscopy, except for the fact that during flagellar movement its distal end remained attached. Thus, the pad is a structure that ensures the adhesion of the nanotubule to the surface, that we will refer to as the "anchoring pad".

### Nanotubules are mainly composed of membrane

To determine the composition of the nanotubules, epimastigotes were stained with CFSE, a cytoplasmic dye, and their membrane with R18, a lipophilic cationic dye derived from rhodamine (see methods) and stretched using a continuous shear of 1.24 pN/$\mu m^2$. Fig 3 shows a

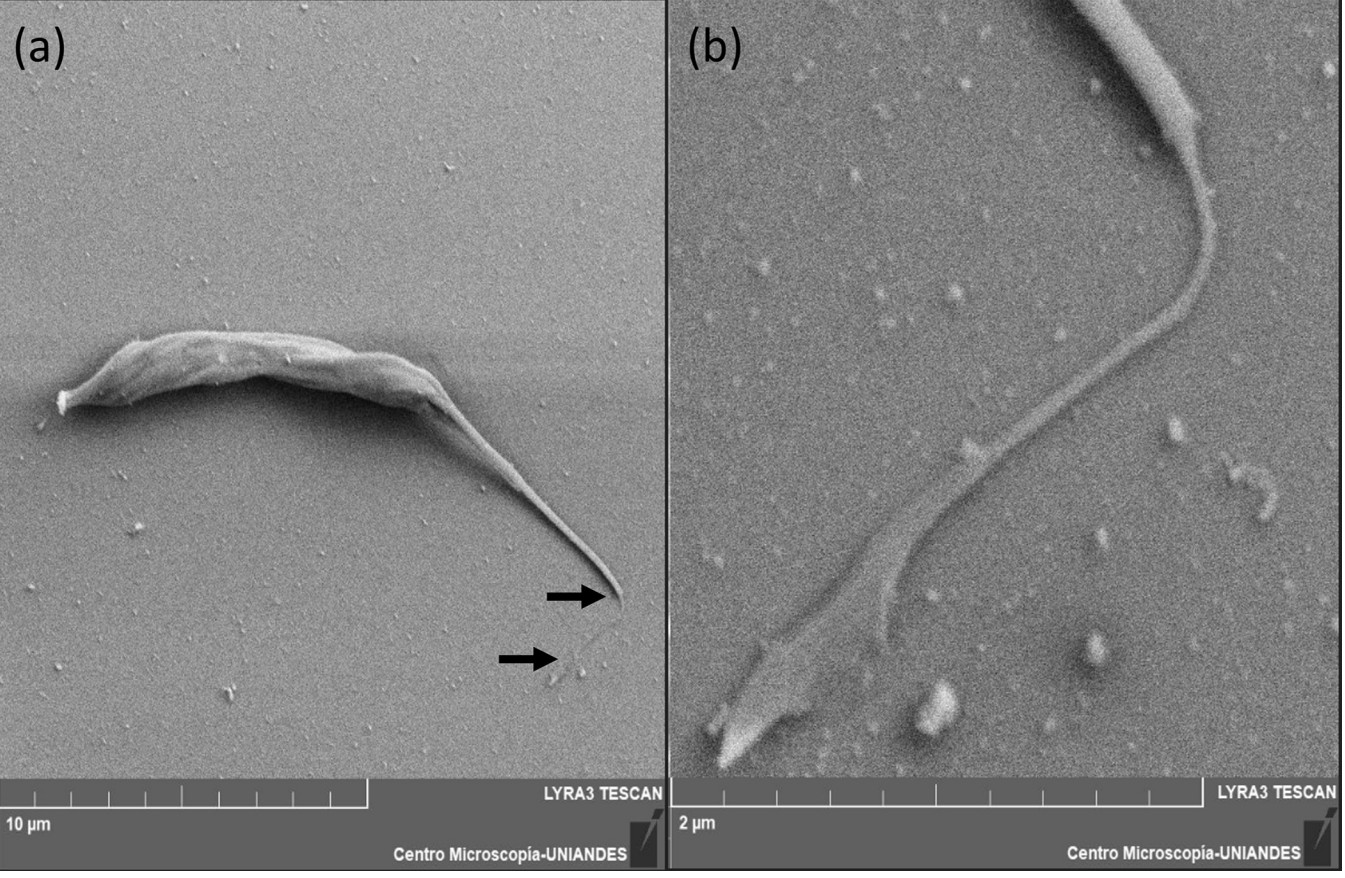

**Fig 2. Electron microscopy images of a shear-stress generated nanotubule.** (a) An epimastigote after being stretched at 1.24 pN/μm². The nanotubule is delimited by black arrows. (b) Detailed view of the nanotubule in (a). Note the anchoring pad at the end of the structure and the funnel shape on the flagellum side.

representative epimastigote after its extension by shear stress. Using bright field microscopy (Fig 3A) it is possible to see the nanotubule by which the epimastigote is adhered to the surface. At the distal end, the anchoring pad appears as a slightly wider structure than the rest of the extended nanotubule. Fig 3B shows CFSE fluorescence mostly at the soma of the parasite. A faint green signal is visible on the nanotubule and the anchoring pad (inset). The signal is not bleed-through from another channel (see the methods section for the description of the negative controls), suggesting that there are few cytoplasmic components along the nanotubule and the anchoring pad. Fig 3C shows R18 fluorescence, where there is a strong signal at the soma of the parasite as well as the anchoring pad, and a lower signal on the nanotubule. Therefore, the nanotubule as well as the pad are mainly composed of membrane, with some proteins. Fig 3D shows the merged CFSE and R18 images, illustrating the differences in distribution with R18 more on the borders and CSFE on the soma of the parasite. Note that difference of fluorescence intensity between the anchoring pad and the nanotubule (~3:1 for CFSE and 5:1 for R18) can be explained by the diffraction limit of the microscope meaning that intensity differences are not necessarily attributable to differences in protein/membrane content between the structures, but to the fact that the light from an area ~40 nm wide is spread over the width of the point spread function of the microscope, which is over 200 nm wide (5:1).

The membrane dye also made it possible to see additional structures that were not seen in brightfield, such as additional nanotubules and attachment points. Fig 4 shows an epimastigote

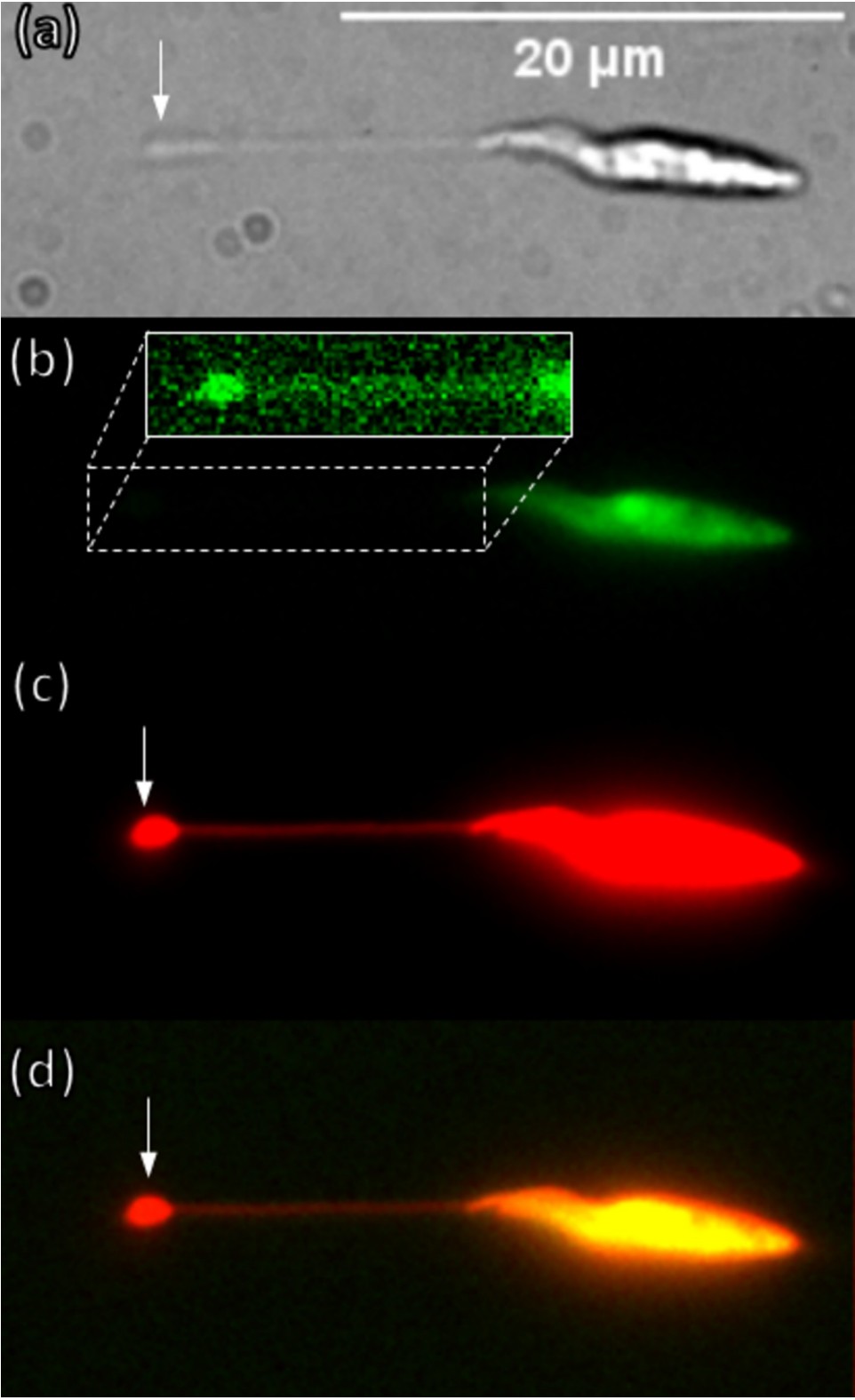

**Fig 3. Nanotubules dyed with cytoplasmic (CFSE) and membrane (R18) dyes.** (a) bright field image of an epimastigote at 90x, after it was extended by shear stress. The anchoring pad is marked by a white arrow at the end of the nanotubule. (b) CFSE fluorescence is mainly at the soma of the parasite, but a faint signal can be seen at both the anchoring pad and the nanotubule (see inset where brightness was increased for visibility) (c) R18 fluorescence levels,

are similar to CFSE fluorescence in the soma, but significantly brighter both in the nanotubule and the anchoring pad. (d) composite of CFSE and R18 images.

at 40X, stretched using a shear stress of 1.55 pN/μm². In bright field (Fig 4A), the presence of nanotubules is not clear, but in fluorescence (Fig 4B), two nanotubules are observed. They have vertices at their distal end, which may correspond to secondary adhesion points formed during the elongation process. Although they appear to be as wide as the nanotubules in the diffraction-limited images, the higher fluorescence intensity in the R18 channel suggests they are larger than the standard nanotubules. This doubly tethered epimastigote was able to resist high flows.

## Nanotubule extension under shear stresses

To determine the dynamics of nanotubules extension as a function of shear stress, parasites were subjected to a cycle of linearly increasing shear stress pulses for 20 s, interspaced by 20 s pauses as shown in Fig 5. When shear stress reached 1.55 pN/μm² most epimastigotes detached from the surface, so measurements were limited to 1.24 pN/μm².

Filaments require a threshold shear stress to extend, and extension increases with shear stress, until detachment. A similar response was observed in most extending epimastigotes (n > 20) and tracked in detail (n = 12). Fig 5A shows the data for one epimastigote; the filament elongates reversibly after each shear stress pulse. Two elongation regimes appear in the extension curves: the filament first extends rapidly in the first ~2–4 s, essentially the rate of change of the pump and hose system, and then creeps slowly, suggesting two mechanisms of elongation. For the rapid elongation interval, the average extension as a function of shear stress (Fig 5B) increases linearly after a threshold. This response to flow is consistent among epimastigotes and the average data of all analyzed epimastigotes, presented in S2 Fig, illustrates the consistency in these responses. Finally, during these 20 s intervals lengths of a few μm up to several 10's μm were observed, with longer lengths for epimastigotes exposed for longer times.

To evaluate the repeatability and the role of previous elongations on the total extension achieved during the extension-contraction cycles, the parasites were subjected to two increasing shear stress cycles with a 1-min pause with no shear stress between the cycles. In the second cycle, the number of epimastigotes extending was reduced from 12 to 5, with 7 strongly attaching to the surface without extending; there was a small reduction, on average, of the maximum elongation reached, as shown in S3 Fig, but the general shape of the elongation curve remained unchanged.

## Multiple tethering improves resistance to high flows

Although most parasites detached from the surface at high shear stress (≥ 1.55 pN/μm²), some remained attached. The epimastigote in Fig 6 was exposed to a shear stress of 1.55 pN/μm² and attached to the surface with two filaments: during the elongation of the first filament, another anchoring point was established, and a second filament extended (S2 Movie). Each filament appears to originate from a different point in the flagellum as shown in Fig 6.

While for flagellum-attached epimastigotes the flagellum wagged vigorously when fluid flow was stopped, for strongly attached epimastigotes movement stopped or was significantly reduced despite the absence of flow, likely due to additional attachment points on the parasite. We observed that this effect was time-dependent: the longer epimastigotes sat on the surface before fluid flow started, the more likely they were to be strongly attached, with most parasites strongly attached after ~3 min without shear stress.

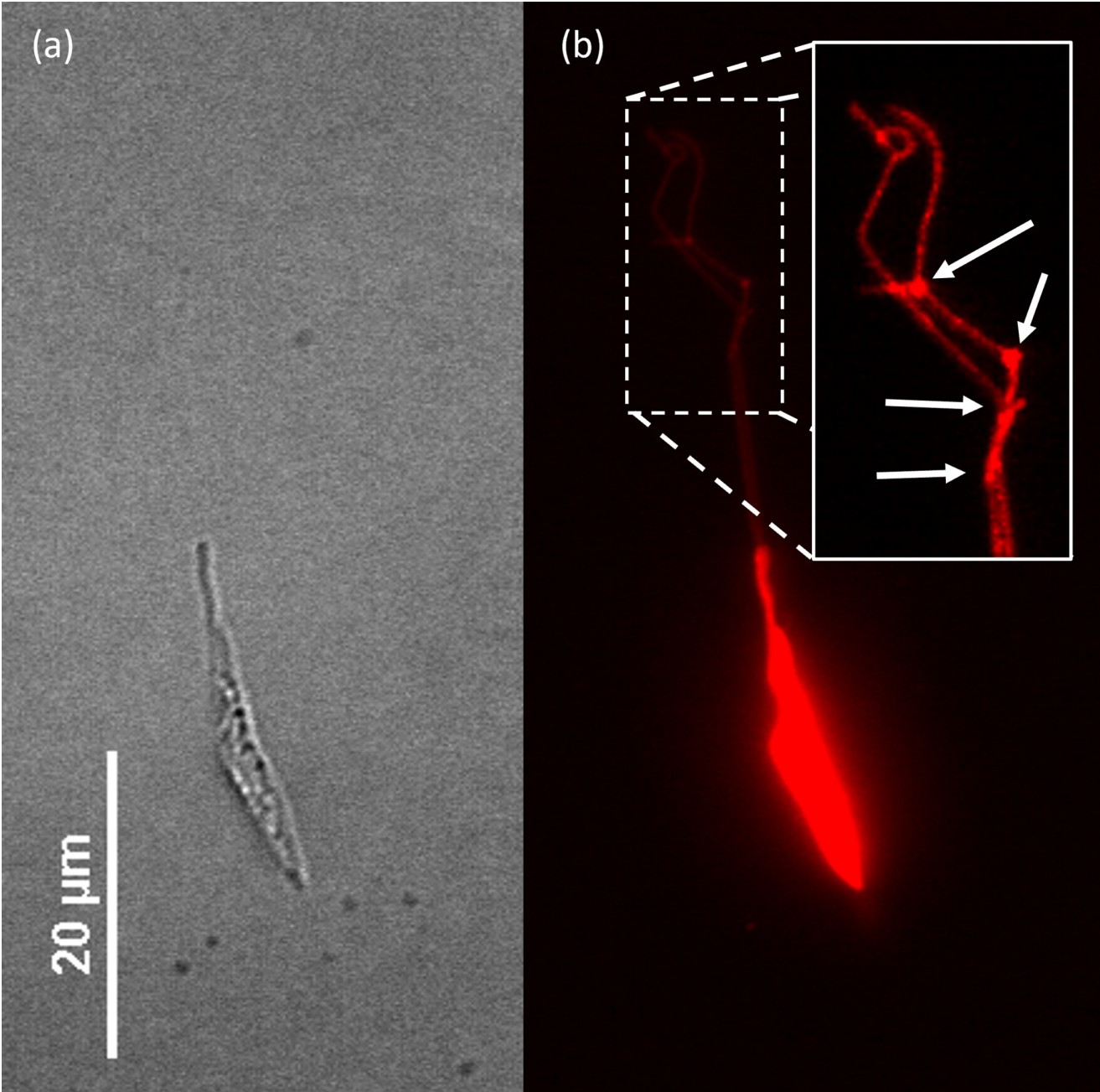

**Fig 4. Secondary adhesion points.** Epimastigote dyed with R18 and subjected to 1.55 pN/μm$^2$ of shear stress; in bright field microscopy at 40X (a) the adhesion structures are not visible, but in fluorescence (b) two nanotubules by which the parasite is attached are visible, as well as secondary anchoring pads (inset: Adjusted brightness, G = 2.89) marked by white arrows.

## The proximal end of nanotubules can migrate

Fig 7 shows the migration of a nanotubule during a cycle of shear stress pulses. The epimastigote was subjected to 0.31 –– 0.62–0.93–1.24 pN/μm$^2$, for 20 s, followed by a 20 s pause. Fig 7A and 7C correspond to the initial state for reference and the pause between the last shear stresses. Flagellar beating in c) induced the rotation of the parasite around its mid-section

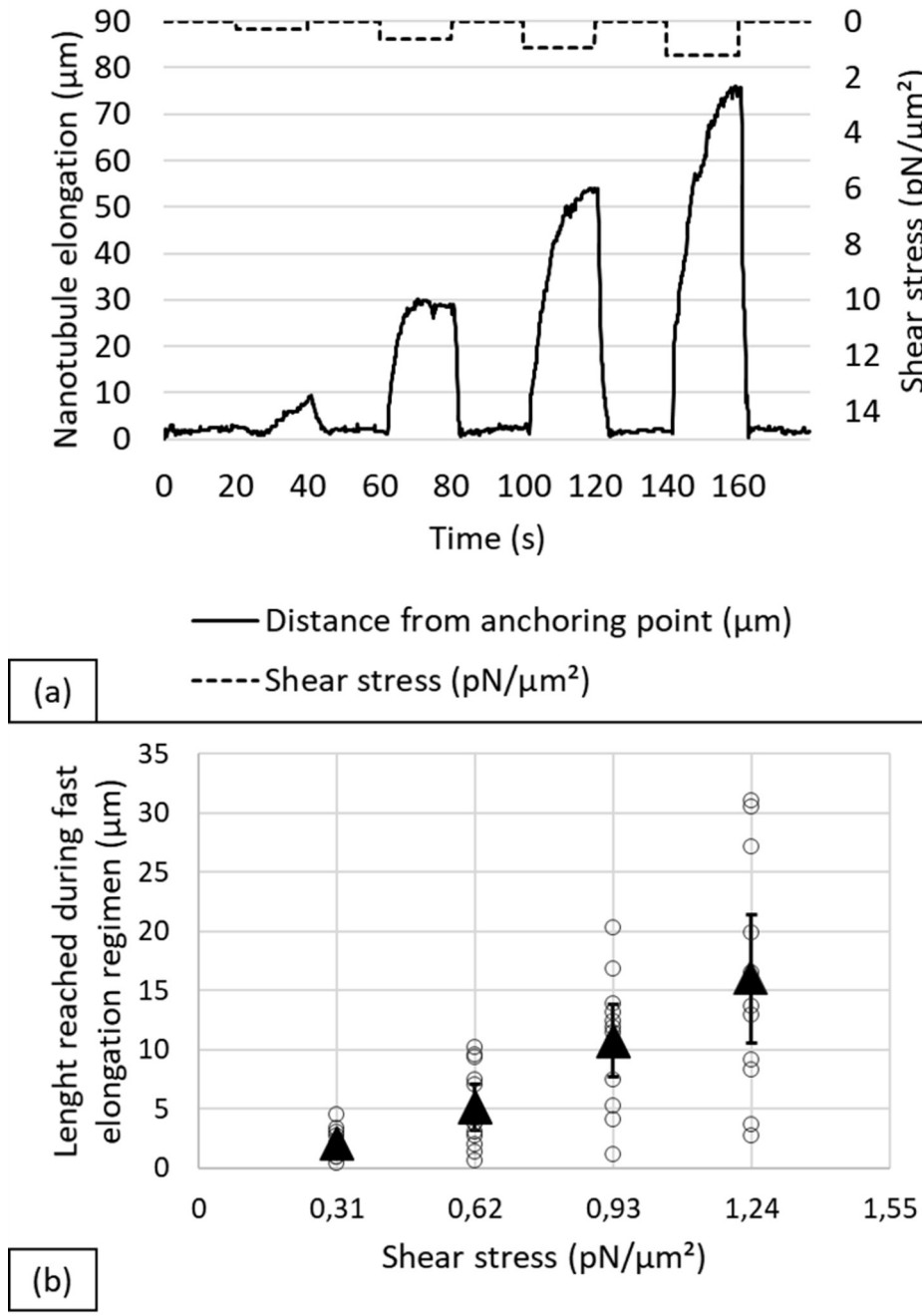

**Fig 5.** (a) Elongation of an epimastigote with increasing shear stress: Starting with 20 secs of no flow, parasites were subjected to a shear stress cycle of 0.31 –– 0.62–0.93–1.24 pN/µm² pulses; each pulse was 20 s long, followed by a pause lasting 20 s. The curve shows the length of the nanotubes, as measured by changes in position of the flagellum tip. (b) Average extension as a function of shear stress. White circles correspond to the average length reached by each parasite during the fast elongation phase (first 2–4 s) after pulse introduction (n = 12); black triangles represent average data (data available in S1 Dataset).

(S3 Movie). When shear stress was applied again at 1.24 pN/µm² (Fig 7D), the parasite acquired a "V" shape with a vertex in the middle of the parasite, indicating the migration of the neck of the nanotubule to a location near the flagellar pocket.

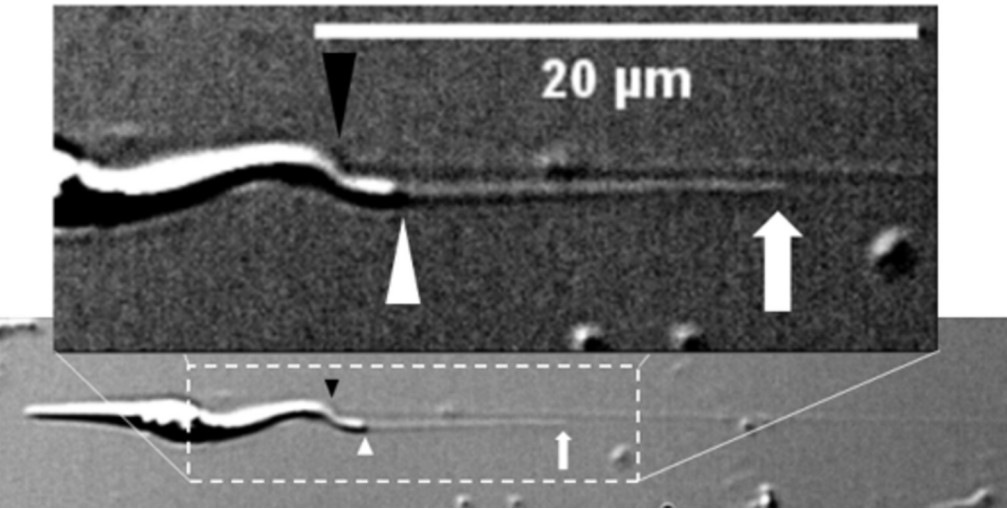

**Fig 6. Doubly tethered epimastigote at 90x magnification.** Inset is a 3X digital magnification of the delimited area. The first filament (on the right) emerged at a shear stress of 1.55 pN/μm$^2$ from an anchoring point established over 100 μm away from the tip of the flagellum (see S4 Fig); During extension, a second anchoring point was established (white arrow), from which the second filament emerged; both extended until the point marked with the black arrowhead for the first filament and the white arrowhead for the second filament, both near the tip of the flagellum.

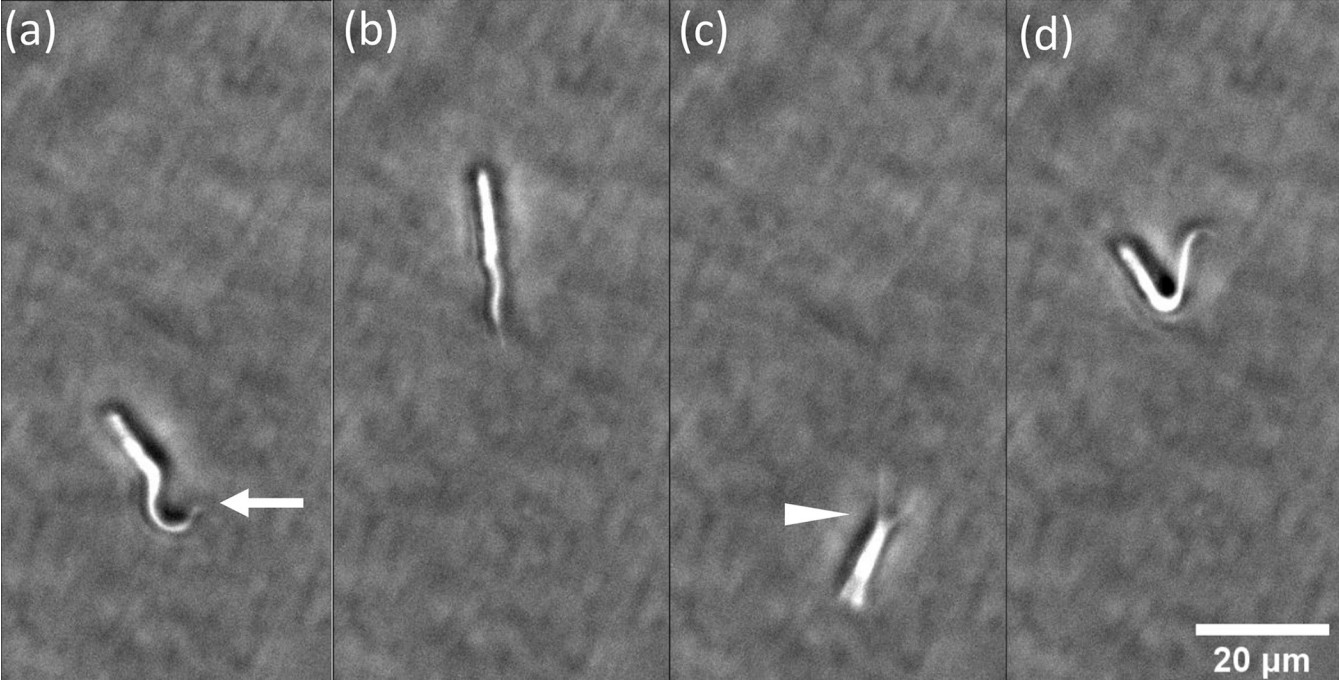

**Fig 7.** Relocation of the parasite-end of a filament during a shear stress cycle with pauses at a) and c). (a) initial adhesion of the parasite to the surface by the anterior part of the flagellum (white arrow) under no flow, (b) after exposure to a shear stress of 0.93 pN/μm$^2$, (c) during the subsequent pause and under 1.24 pN/μm$^2$. Flagellar beating during the pause resulted in the rotation of the parasite around a pivot point marked by the white arrowhead and near the flagellar pocket. At 1.24 pN/μm$^2$, the parasite acquires a "V" shape shown in (d), indicative of the change in the position of the parasite-end of the nanotubule.

## Discussion

Trypomastigotes travel in the bloodstream of mammals, and in humans they invade the heart. The same occurs for epimastigotes, which must travel along the intestine of the vector until it adheres to an enterocyte to initiate metacyclogenesis; however, little is known about the process of adhesion and invasion under flow. Mechanisms that overcome high fluid flow and make it possible for pathogens to remain attached to the surface have previously described in different pathogens [26–35]. Here, we found a similar mechanism: In epimastigotes attached to the surface, shear stress induces the formation of nanotubes, mainly composed of membrane and a few proteins. The length of nanotubes increases with shear stress and can result in nanotubes up to 100 μm long. At the distal end, attachment is mediated by an anchoring pad (Fig 3), which has a higher protein content in comparison to the filaments, likely proteins associated with surface adhesion.

Similar membrane tether formation processes have been described for membrane tethers in erythrocytes [36], neutrophils [37–39], fibroblasts [40], *Escherichia coli* [41], among others. Characteristically, when fluid flow stops, the nanotubule is reabsorbed and the parasite returns to its original position, implying a reversible process when they are attached by a single tether. A threshold force was necessary to initiate the extraction of membrane tethers, as documented in other systems [42–46], which has been interpreted as the force required to overcome membrane-cytoskeleton interactions [47], and implying the presence of interactions between the membrane and the flagellum. Here, we estimate the force threshold required for nanotubule extrusion at ~7 pN from the intercept of the rapid elongation phase presented in Fig 5B (see methods). This is consistent with the forces required in other types of cells [48]. Additionally, since models for force-extruded tethers or nanotubules relate their radius to the curvature modulus [24,25], using the tether radius obtained in SEM and the force at which the nanotubule was extruded, a curvature modulus of ~60 $k_bT$ was estimated for the membrane (see methods), in line with that of other types of cells [25]. It is also possible to estimate the surface viscosity from a viscous drag model [25] using the elongation velocity vs force curve (S1 Fig) at $50 \, 10^{-6} \, Pa \cdot s \cdot m$, also consistent with the properties of the membranes of other types of cells [25].

The dynamics of elongation shows a fast extension (first 2–4 s) followed by creep. Similar elongation processes have been reported in membrane tethers by force experiments, where the initial phase corresponds to the elastic component of the membrane, while the slow regimen can be attributed to a gradual addition of phospholipids to the tether from membrane reservoirs [36,40,41,45,46]. Additionally, the reduction in activity during elongation or multiple tethering (which likely maintains tension as in strongly attached bacteria [30]) is suggestive of a mechanism to sense membrane tension that could establish when the parasite is firmly adhered in order to continue its next step of the invasion process.

Epimastigotes that were attached by multiple nanotubules remained attached to the surface at higher shear stress magnitudes than those singly attached. This is consistent with simulations of *E. coli* attachment via multiple tethers, where increased attachment at high and variable flows is promoted by the distribution of force between tethers with yielding elasticity [30], meaning that force plateaus in the extension curve, as is seen here during the slow extension regime. This is also observed experimentally for membrane tethers extracted mechanically in Chinese hamster ovary and a malignant human brain tumor, and endothelial cells by Sun *et al.* [42]: individual tethers contribute to increase the total pulling force. Thus, tethers, which have also been related to buffering the force from fluid flow [30,45,49–51], could regulate adhesion under stresses generated in the gut of the triatomine host as can be the case for bacteria in the human gut [52].

Nanotubules can emerge or migrate to parts other than the flagellum tip such as the flagellar pocket. Migration of protein structures on the membrane has been observed in *Trypanosoma brucei*, the agent of African trypanosomiasis. Engstler *et al.* described a process involving the clearance of antibodies and immune complexes on the surface of the parasite, where anti-glycoproteins antibodies were mobilized to the flagellar pocket zone for endocytosis and posteriorly eliminated [10]. More experiments are needed to establish the role (if any) of nanotubule migration in these parasites, and a possibility could improve the distribution of force in multiply attached cells.

Other membrane/cytoplasmic projections, known as filopodia, have been described in *T. cruzi* [53], and also in other parasites from the Trypanosomatida order. In *T. brucei* adhesion and antibody clearance have been proposed [54,55]. Conversely, Szempruch *et al.* [56] induced the formation of tubular structures from the flagellar membrane of *T. brucei*. These structures did not result from mechanical stress and were composed of flagellar membrane proteins as well as virulence factors that could be transferred to other cells by direct contact [57] or by vesicles resulting from the subsequent fragmentation of the nanotubules [56].

To our knowledge, the formation of similar structures due to forces from fluid flow have not been previously reported in *T. cruzi*. The nanotubules described here have characteristics that can facilitate the adhesion process of a flagellated form of the parasite under flow, necessary for replication or maturation inside the host. Membrane-derived trails have been described in *T. cruzi* amastigotes and epimastigotes [58]. These appeared to emerge spontaneously from the flagellar pocket or the opposite side of the parasites, in a time and temperature dependent manner, in both *in-vitro* and *in-vivo* conditions. Although the formation process of the trails was not entirely clear, given that there are similarities with the trails in Fig 4, it is possible that these structures resulted from the flow forces of the washing step to remove unbound parasites in the protocol because the formation was independent of the metabolic conditions of amastigotes. Alternatively, they could be part of a different process because they were beaded in appearance (possibly due to microvesicles as in [59]) and were mainly studied in amastigotes since they were hardly detected in epimastigotes.

A significant difference with respect to membrane tethers observed in other cells is that here tethers mediate strong attachment to a surface, whereas in most other systems this role is not observed, and in neutrophils they mediate transient attachment during rolling [39]. Moreover, the anchor points at the distal end of tethers in neutrophils also appear to be smaller than the anchoring pads observed here [39,60] indicating a different role of the nanotubules in neutrophils.

## Conclusions

We showed shear stress can induce nanotubular projections from the flagellar membrane of *T. cruzi* epimastigotes after surface attachment via an anchoring pad on the surface. Those structures are mainly composed of membrane, as suggested from fluorescence staining, and by their mechanical properties such as extrusion force, curvature modulus, surface viscosity and a yielding force-extension curve. These membrane tethers can extend significantly, up to 100µm, and migrate along the flagellar membrane. Extension is reversible, unless the parasite can form additional attachments, in which case resistance to high fluid forces is greatly enhanced. Although further work is needed to improve our understanding of the biophysical and biochemical properties as well as the biological significance of the nanotubules in the life cycle of epimastigotes, these results suggest a function for membrane nanotubules in *T. cruzi*: to mediate adhesion under flow inside the microenvironment of the host. In the wider context of cell systems under flow, a novel function for membrane tethers is shown, that of strong

adhesion under high shear stresses different from that of transient adhesion, as observed in neutrophils.

## Supporting information

You can find the supporting information at https://doi.org/10.6084/m9.figshare.22178795.

**S1 Fig. Nanotubule elongation velocity vs force.** The line is a fit using the model by Bro-chard-Wyart et al. [25] and results in an estimate of surface viscosity of the flagellar membrane of *Trypanosoma cruzi* at ~50 $10^{-6}$ *Pa·s·m*, consistent with the properties of cell membranes from other types of cells.
(TIF)

**S2 Fig. Average length reached by 12 epimastigotes along a first cycle.** Each epimastigote was subjected to a shear stress cycle of 0.31 –– 0.62–0.93–1.24 pN/um$^2$; each pulse was 20 s long and followed by a 20 s pause. There was an initial pause 20 s long. The curve shows the average elongation of 12 different parasites. Note both elongation regimes: a fast extension and a creep regime. Differences between epimastigotes were mainly related to the maximum length reached at each shear stress.
(TIF)

**S3 Fig. Average length reached by 5 epimastigotes in two shear stress cycles.** Each epimasti-gote was subjected to two shear stress cycles of 0.31 –– 0.62–0.93–1.24 pN/um$^2$; each pulse was 20 s long and followed by a 20 s pause. There was a pause of 1 min between cycles. The fila-ment length during this pause was averaged and set to 0, assuming this as the initial anchoring point for the second cycle. The curve shows the average elongation of five different parasites. There is a subtle reduction in maximum length reached in the second cycle.
(TIF)

**S4 Fig. Doubly tethered epimastigote.** Illustrative multiply tethered epimastigote, observed at a shear stress of 1.55 pN/μm$^2$. The anchoring point of the first adhered filament is marked with the black arrow. Note how far the filament extends without rupturing or detaching. The second filament emerged during the extension of the first one, its anchoring point is marked by a white arrow. The inset shows the parasite-end of the filaments (black arrow head for the first filament and white arrowheat for the second), note the diferent location of each one.
(TIF)

**S1 Table. Flow of PBS used, and its shear stress equivalences.**
(XLSX)

**S1 Dataset. Extension data and calculations.** https://doi.org/10.6084/m9.figshare.22178789.v1.
(XLSX)

**S1 Movie. Nanotubule extension affects flagellar beating.** While subjected to a shear stress of 0.62 pN/μm$^2$, a tether-like structure emerges from the flagellum of this surface-bound epi-mastigote and beating is significantly reduced. Then, when flow is stopped, the flagellum beat-ing resumes while the filament is still attached. https://doi.org/10.6084/m9.figshare.22177418.v1.
(MP4)

**S2 Movie. Emergence of a secondary filament.** An epimastigote exposed to a shear stress of 1.55 pN/μm$^2$ is attached to the surface by a single filament. During the elongation of this fila-ment, a new anchoring point is established, and a second filament emerges. https://doi.org/10.

6084/m9.figshare.22177583.v1.
(MP4)

**S3 Movie. Different patterns of extension suggest nanotubule migration.** An epimastigote is exposed to a shear stress of 0.93 pN/$\mu$m$^2$. Then flow is stopped, and the parasite returns to its initial position where flagellar beating induces the rotation of the parasite. When flow is turned on again to a shear stress of 1.24 pN/$\mu$m$^2$, it extends and acquires a "V" shape, with the vertex located near the flagellar pocket. This observation suggests that the nanotubule migrated near the flagellar pocket. https://doi.org/10.6084/m9.figshare.22177586.v1.
(MP4)

## Acknowledgments

We acknowledge the technical assistance of the $\mu$-Core, as well as Elizabeth Suesca for her support in the lab, and Jaime Castro for early experiments.

## Author Contributions

**Conceptualization:** Cristhian David Perdomo-Gómez, Nancy E. Ruiz-Uribe, John Mario González, Manu Forero-Shelton.

**Formal analysis:** Cristhian David Perdomo-Gómez, Manu Forero-Shelton.

**Funding acquisition:** Manu Forero-Shelton.

**Investigation:** Cristhian David Perdomo-Gómez, Nancy E. Ruiz-Uribe.

**Methodology:** Cristhian David Perdomo-Gómez, Nancy E. Ruiz-Uribe, John Mario González, Manu Forero-Shelton.

**Project administration:** Manu Forero-Shelton.

**Resources:** John Mario González, Manu Forero-Shelton.

**Supervision:** John Mario González, Manu Forero-Shelton.

**Validation:** Manu Forero-Shelton.

**Visualization:** Cristhian David Perdomo-Gómez.

**Writing – original draft:** Cristhian David Perdomo-Gómez.

**Writing – review & editing:** Nancy E. Ruiz-Uribe, John Mario González, Manu Forero-Shelton.

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
