## [Decision Letter · Decision Letter 0]

11 Jan 2023

PONE-D-22-32336Extensible membrane nanotubules mediate attachment of Trypanosoma cruzi epimastigotes under flowPLOS ONE

Dear Dr. Forero-Shelton,

Thank you for submitting your manuscript to PLOS ONE. After careful consideration, we feel that it has merit but does not fully meet PLOS ONE’s publication criteria as it currently stands. Therefore, we invite you to submit a revised version of the manuscript that addresses the points raised during the review process.

We look forward to receiving your revised manuscript.

Kind regards,

Vyacheslav Yurchenko, Ph.D.

Academic Editor

PLOS ONE

Journal Requirements:

Additional Editor Comments:

The work of Perdomo-Gómez et al. on membrane nanotubules-mediated attachment of Trypanosoma cruzi epimastigotes under flow was reviewed by three independent scientists and they all were positive and supportive of its publication in PLoS One. Please address their comments and submit a revised manuscript, which (likely) will not go through another round of peer-review.

Reviewers' comments:

Reviewer's Responses to Questions

**Comments to the Author**

1. Is the manuscript technically sound, and do the data support the conclusions?

Reviewer #1: Yes

Reviewer #2: Yes

Reviewer #3: Yes

2. Has the statistical analysis been performed appropriately and rigorously? 

Reviewer #1: N/A

Reviewer #2: Yes

Reviewer #3: N/A

3. Have the authors made all data underlying the findings in their manuscript fully available?

Reviewer #1: Yes

Reviewer #2: Yes

Reviewer #3: Yes

4. Is the manuscript presented in an intelligible fashion and written in standard English?

Reviewer #1: Yes

Reviewer #2: Yes

Reviewer #3: Yes

5. Review Comments to the Author

Reviewer #1: This is a very interesting and original manuscript focusing on how epimastigote forms may attach through their flagelum. Those forms are usually found in the triatomine vector where parasites should attach during their life cyclo to avoid passage with the blood meal.The authors showed using microscopic (SEM and Immunofluorescence) and videos a mechanism of in vitro attachment that open perspective to undertande flagellar extension using nanotubules. The paper is very well-written and I found only some misspellings. I therefore recommend acceptance of the manuscript.

Major comments

I think the text (lines 1-55) could be shortened or even removed since it includes basic information on T. cruzi biology. Introduction could start at the 3rd paragraph since the flagellum is the main focus of the paper.

Line 425 - I do not think is the GPI-mucins, it is probably another major glycoconjugate, the GIPLs. The best reference to cite here is: Trypanosoma cruzi: involvement of glycoinositolphospholipids in the attachment to the luminal midgut surface of Rhodnius prolixus.

Nogueira NF, Gonzalez MS, Gomes JE, de Souza W, Garcia ES, Azambuja P, Nohara LL, Almeida IC, Zingales B, Colli W.

Exp Parasitol. 2007 Jun;116(2):120-8. doi: 10.1016/j.exppara.2006.12.014.

Minor comments.

line 195 - nanotubules

lines 325, 336/337 and 370/399 - 1,55/1,24 or 1.55/1.24?

Lines 361-362/396 and several parts in figure legends - 0.31 –– 0.62 – 0.93 – 1.24

line 460 - total?

line 473 - brucei not Brucei

Reviewer #2: The submitted manuscript has been transferred from Plos Pathogens following revisions to address the main comments of the reviewers. The revised manuscript is well written and it adequately addresses the concerns regarding the relevance of the filaments to attachment in vivo, and the reproducibility of the observations regarding filament elongation under varying levels of shear stress. Despite the absence of in vivo studies, the data on on the biophysical properties of the filaments are sound, they are novel with respect to T. cruzi, and provide a meaningful advance to the study of microbial adhesion under conditions of shear stress.

Reviewer #3: I enjoyed so much reading this manuscript. It is shown that epimastigotes submitted to shear force after attachment to a surface release nanotubes that is formed by membranes of the parasite. I have some comments that could help to clarify some issues I found.

1- it should be discussed that attachment only occurs in parasite washed in PBS (why?) and extended in PBS.

2- The nanotubes can be seen after addition of medium and shear force (I know that they remain attached)?

3- The authors should clarify what surface they used? Only glass?

4- The text should include and discuss the formation of similar structures in T. cruzi amastigotes (J . Euk. Microbioi.,43(4)1996, 275-28 and references in).

5- The author suggest that T. cruzi mucins could be in the membrane of the nanotubes. It would be interesting if they test it using monoclonal antibodies. There are also antibodies that labeled the flagellum membrane (FCBP for example) that could be used, if possible.

6- Recheck orthography. I found a few typos.

6. PLOS authors have the option to publish the peer review history of their article (what does this mean?). If published, this will include your full peer review and any attached files.

Reviewer #1: **Yes: **Rodrigo Pedro Soares

Reviewer #2: **Yes: **David Sacks

Reviewer #3: **Yes: **Sergio Schenkman

---

## [Author Response · Author response to Decision Letter 0]

2 Mar 2023

1. Please ensure that your manuscript meets PLOS ONE's style requirements, including those for file naming. The PLOS ONE style templates can be found at https://journals.plos.org/plosone/s/file?id=wjVg/PLOSOne_formatting_sample_main_body.pdf and https://journals.plos.org/plosone/s/file?id=ba62/PLOSOne_formatting_sample_title_authors_affiliations.pdf.

Response: We have revised our manuscript in order to fulfill the journal’s requirements

Response: The funding we received was not in the form of formal grants, and there are no grant numbers. We included the (intramural) financial support in the acknowledgements.

b) If there are no restrictions, please upload the minimal anonymized data set necessary to replicate your study findings as either Supporting Information files or to a stable, public repository and provide us with the relevant URLs, DOIs, or accession numbers. For a list of acceptable repositories, please see http://journals.plos.org/plosone/s/data-availability#loc-recommended-repositories.te

Response: we have uploaded the data which we analyzed as a dataset and the movies to Figshare. You can find them in: 

supporting information: https://doi.org/10.6084/m9.figshare.22178795

S1 Dataset: https://doi.org/10.6084/m9.figshare.22178789.v1

S1 Movie: https://doi.org/10.6084/m9.figshare.22177418.v1

S2 Movie: https://doi.org/10.6084/m9.figshare.22177583.v1

S3 Movie: https://doi.org/10.6084/m9.figshare.22177586.v1

Response: In that particular section we are describing the biophysical properties of adhesion nanotubules in T. cruzi epimastigotes after they were exposed to shear stress. We observed that some epimastigotes did not produce nanotubules, so we did not consider those parasites for the analysis of nanotubule properties, even if we observed them, so the phrase “data not shown” was inaccurate. We have rewritten lines 379 – 385 to clarify that.

Response: 

Concerning the articles we referenced in our manuscript, we could not find retracted articles. We did find one paper that has a corrected section that is not related to our study or the data we cited. Nevertheless, taking into consideration a comment given by one of the reviewers (below), we have shortened the first part of the introduction, the section in which we used the mentioned article, and we have removed that reference from our manuscript.

On the other hand, we have added references 8 and 9 to broaden the description of the flagellum functionalities in the introduction. 

(Removed) Ballesteros-Rodea G, Santillán M, Martínez-Calvillo S, Manning-Cela R. Flagellar Motility of Trypanosoma cruzi Epimastigotes. Journal of Biomedicine and Biotechnology. 2012 

8. Vaughan S. Assembly of the flagellum and its role in cell morphogenesis in Trypanosoma brucei. Curr Opin Microbiol. 2010 Aug;13(4):453–8. 

9. Sharma R, Peacock L, Gluenz E, Gull K, Gibson W, Carrington M. Asymmetric Cell Division as a Route to Reduction in Cell Length and Change in Cell Morphology in Trypanosomes. Protist. 2008 Jan;159(1):137–51.

Response to de reviewers

• Reviewer #1, Dr. Rodrigo Pedro Soares

1. I think the text (lines 1-55) could be shortened or even removed since it includes basic information on T. cruzi biology. Introduction could start at the 3rd paragraph since the flagellum is the main focus of the paper. 

Response: We thank the reviewer for the comment, and agree the introduction was a bit lengthy. We have rewritten the introduction to improve its focus and clarity.

2. Line 425 - I do not think is the GPI-mucins, it is probably another major glycoconjugate, the GIPLs. The best reference to cite here is: Trypanosoma cruzi: involvement of glycoinositolphospholipids in the attachment to the luminal midgut surface of Rhodnius prolixus.

Nogueira NF, Gonzalez MS, Gomes JE, de Souza W, Garcia ES, Azambuja P, Nohara LL, Almeida IC, Zingales B, Colli W.

Exp Parasitol. 2007 Jun;116(2):120-8. doi: 10.1016/j.exppara.2006.12.014.

Response: We agree that the proteins involved in the process could indeed be glycoinositolphospholipids (GIPL) and the article suggested is very interesting; considering that the focus of the manuscript is on the biomechanical properties of membrane nanotubules, we have removed assumptions regarding the molecules that could mediate attachment. 

Minor comments.

a. line 195 -– nanotubules Corrected

b. lines 325, 336/337 and 370/399 - 1,55/1,24 or 1.55/1.24? We corrected all numbers to period separated decimals instead of comma separated decimals.

c. Lines 361-362/396 and several parts in figure legends - 0.31 –– 0.62 – 0.93 – 1.24 corrected

d. line 460 - total? Corrected

e. line 473 - brucei not Brucei ✓ Corrected

• Reviewer #2, Dr. David Sacks

o The submitted manuscript has been transferred from Plos Pathogens following revisions to address the main comments of the reviewers. The revised manuscript is well written and it adequately addresses the concerns regarding the relevance of the filaments to attachment in vivo, and the reproducibility of the observations regarding filament elongation under varying levels of shear stress. Despite the absence of in vivo studies, the data on on the biophysical properties of the filaments are sound, they are novel with respect to T. cruzi, and provide a meaningful advance to the study of microbial adhesion under conditions of shear stress.

Response: We appreciate the time taken to review and comment the paper.

• Reviewer #3, Dr. Sergio Schenkman

1. It should be discussed that attachment only occurs in parasite washed in PBS (why?) and extended in PBS.

Response: We thank the reviewer for the comment. We used PBS because it has a known and stable viscosity (necessary to calculate shear stress and the resulting forces on the parasite), is isoosmolar; then, we did not disturb the homeostasis or motility of the parasite. Attachment of the parasites was only studied under PBS, but is likely to happen in other buffers.

2. The nanotubes can be seen after addition of medium and shear force (I know that they remain attached)?

Response: Parasites were added in the medium, and nanotubules were observed after these attached to the surface and force resulting from shear stress was turned on. We did not change media for the reasons explained above.

3. The authors should clarify what surface they used? Only glass?

Response: We used non-functionalized glass, and have clarified this in the methods, we thank the reviewer for pointing this oversight. We also tested poly-l-lysine but found it was not necessary to study the mechanical properties of the nanotubules and resulted in additional complexity.

4. The text should include and discuss the formation of similar structures in T. cruzi amastigotes (J . Euk. Microbioi.,43(4)1996, 275-28 and references in).

Response: We thank the reviewer for pointing it out this paper as we had missed it. We now discuss it in the context of our findings in lines 512 - 526. Similarly, we included reference 59 to enrich the discussion.

5. The author suggest that T. cruzi mucins could be in the membrane of the nanotubes. It would be interesting if they test it using monoclonal antibodies. There are also antibodies that labeled the flagellum membrane (FCBP for example) that could be used, if possible.

Response: We agree that this would be an interesting experiment to attempt in order to elucidate the molecules involved in this attachment process. However, considering that the scope of our research is to describe the biomechanical properties of the nanotubules in the attachment process, not to identify the molecules that mediate attachment, we removed the part where we discuss the molecules that could mediate the interaction since it is detract from the focus of the paper.

6. Recheck orthography. I found a few typos

Response: We corrected a couple typos we found using our text editing software, hopefully they are the same.

Again, we thank the reviewers and the editor for taking the time to review the paper and make constructive comments that have improved the quality of the paper.

---

## [Editor Report · Decision Letter 1]

6 Mar 2023

Extensible membrane nanotubules mediate attachment of Trypanosoma cruzi epimastigotes under flow

PONE-D-22-32336R1

Dear Dr. Forero-Shelton,

We’re pleased to inform you that your manuscript has been judged scientifically suitable for publication and will be formally accepted for publication once it meets all outstanding technical requirements.

Kind regards,

Vyacheslav Yurchenko, Ph.D.

Academic Editor

PLOS ONE

Additional Editor Comments (optional):

Kudos to authors for producing an interesting and important paper!
---

## [Editor Report · Acceptance letter]

14 Mar 2023

PONE-D-22-32336R1 

Extensible membrane nanotubules mediate attachment of *Trypanosoma cruzi* epimastigotes under flow 

Dear Dr. Forero-Shelton:

I'm pleased to inform you that your manuscript has been deemed suitable for publication in PLOS ONE. Congratulations! Your manuscript is now with our production department. 

Kind regards, 

on behalf of

Prof. Vyacheslav Yurchenko 

Academic Editor

PLOS ONE